# Implications of GPIIB-IIIA Integrin and Liver X Receptor in Platelet-Induced Compression of Ovarian Cancer Multi-Cellular Spheroids

**DOI:** 10.3390/cancers16203533

**Published:** 2024-10-19

**Authors:** Zitha Redempta Isingizwe, Virginie Sjoelund, Doris Mangiaracina Benbrook

**Affiliations:** 1Department of Pharmaceutical Sciences, College of Pharmacy, University of Oklahoma Health Sciences Center, Oklahoma City, OK 73117, USA; zitha-isingizwe@ouhsc.edu; 2Department of Biochemistry, College of Medicine, University of Oklahoma Health Sciences Center, Oklahoma City, OK 73104, USA; v.sjoelund@northeastern.edu; 3Division of Gynecologic Oncology, Department of Obstetrics and Gynecology, College of Medicine, Stephenson Cancer Center, University of Oklahoma Health Sciences Center, Oklahoma City, OK 73104, USA

**Keywords:** ovarian cancer, spheroids, platelets, platelet inhibitors, liver X receptor, integrin signaling, eptifibatide

## Abstract

**Simple Summary:**

Patients with ovarian cancer often have higher levels of platelets in their blood than are normally found in healthy individuals. Ovarian cancer cells form clumps called spheroids that travel throughout the abdomen in a fluid that contains platelets. Previously, we showed that platelets cause ovarian cancer spheroids to become smaller and denser. In this study, we found that an antiplatelet inhibitor called eptifibatide can prevent this effect of platelets on ovarian cancer spheroids. We also discovered that the molecule called liver X receptor appears to be involved in how platelets affect ovarian cancer spheroids and how eptifibatide can prevent this. These findings suggest that eptifibatide could be repurposed to treat ovarian cancer, and that drugs that affect the liver X receptor could be developed to enhance the anti-cancer activity of eptifibatide.

**Abstract:**

**Background:** Platelets have been shown to promote ovarian cancer; however, the mechanism is poorly understood. Previously, we demonstrated that platelets reduce the size and increase the density of multi-cellular ovarian cancer spheroids in cell cultures. The objectives of this study were to determine if platelet inhibitors could counteract these effects, and to explore the mechanisms involved. **Methods:** FDA-approved platelet inhibitors were screened for their abilities to alter platelet effects on ovarian cancer spheroids. Mass spectrometry was used to identify proteins significantly altered in cancer cells upon exposure to platelets. The effects of platelets and/or liver x receptor agonists or antagonists on LXR activity were measured using ES-2 ovarian cancer cells transduced with an LXR-reporter vector. **Results:** Eptifibatide, a GPIIB-IIIA integrin inhibitor, and dipyridamole, an adenosine reuptake inhibitor, reduced and enhanced platelet effects on ovarian cancer spheroids, respectively. Proteomic studies identified the LXR/RXR and integrin pathways as mediators of platelet effects on ovarian cancer, and downstream effectors of eptifibatide. **Conclusions:** Integrin pathways and their downstream LXR/RXR effectors are implicated in how platelets alter ovarian cancer spheroid morphology. These results support studying eptifibatide and LXR/RXR agonists as candidate drugs for repurposing as therapeutic strategies to counteract platelet promotion of ovarian cancer.

## 1. Introduction

While the predicted number of new ovarian cancer cases in 2024 is only about 19,710 in the US, the predicted number of deaths due to the disease during the same period is 13,210, making ovarian cancer one of the most lethal cancers. It is the fifth-leading cause of cancer related deaths in women, and contributes a total of 4% of all cancer deaths in women [1]. At its initial stages, ovarian cancer causes generalized everyday symptoms that are usually dismissed, such as abdominal bloating or swelling, constipation, appetite loss, weight loss, and discomfort in the pelvic area. There are multiple origins of ovarian cancer, including the ovarian epithelial surface and fallopian tube fimbriae [2,3]. More than 90% of the cases are epithelial, and the high grade serous-type histology occurs at the highest incidence (>70%) [4]. More than two-thirds of ovarian cancer patients are diagnosed at stage III, where there is involvement of the upper abdomen and lymph nodes, and stage IV, where there is a distant metastatic disease such as pleural effusion [5]. This late diagnosis complicates the current course of treatment, which is debulking surgery combined with taxane/platinum-based chemotherapy. In fact, while the five-year survival rate before metastasis is 92%, this rate goes down to 42% and 26% for stages III and IV, respectively [6].

The later stages of ovarian cancer are associated with thrombocytosis, defined as platelet counts above 450,000/µL of blood, which is predictive of poor survival [7]. Platelets are rich in growth factors such as vascular endothelial growth factor, which is the main driver of tumor angiogenesis and the formation of new blood vessels. Patients with thrombocytosis have higher resistance to chemotherapy, and for chemotherapy-resistant patients, their platelet count does not normalize with treatment [8]. A high thrombosis risk and a high risk of developing resistance to chemotherapy in cancer patients who exhibit thrombocytosis has also been reported [9,10].

Platelets play a driving role both in vitro and in vivo to promote tumor metastasis, the major cause of ovarian cancer deaths. Cancer cells activate and aggregate platelets, while platelets promote metastasis by physically protecting cancer against natural killer cells, enhancing attachment to the blood vessels and disrupting endothelial junctions [11]. A direct contact between platelets and cancer cells has been shown to promote extravasation of platelets [11]. An increase in the number of platelet receptors, such as P-selectin, has been reported in cancer compared to healthy cells [12]. Platelets can infiltrate tumors to promote angiogenesis through selective sequestering, transport, and release of growth factors, which have direct effects on tumor growth [7,11]. For example, transforming growth factor β (TGFβ) produced by platelets activates the epithelial-to-mesenchymal transition through the TGFβ/suppressor of mothers against decapentaplegic (SMAD) and nuclear factor-κB (NF-κB) pathways, which have direct effects on tumor growth [11]. In co-cultures of platelets and cancer cells, platelets have been shown to induce the mesenchymal phenotype of cancer cells, as characterized by upregulation of Twist1 and reductions in E-cadherin, which in turn enhanced cancer cell motility and platelet pro-aggregation status [13]. In the same study, the injection of platelets into tumor-bearing mice increased lung metastasis as compared to non-platelet injected mice, indicating that platelets had the ability to promote tumor metastasis. Furthermore, pro-platelet aggregation states have been reported for cancer patients, relating to platelet release of their granule contents [12].

We developed a 3D magnetic ovarian cancer spheroid-platelet co-culture model to test platelet-mediated effects on ovarian cancer cells grown as multi-cellular spheroids. Using this model, we demonstrated that platelets altered the morphology of ovarian cancer spheroids, causing them to become smaller and denser spheroids, and ruled out the roles of dihomo-gamma-linolenic acid in the mechanism [14]. Magnetic compression of colorectal cancer spheroids prior to injection in mice was shown to be associated with a more aggressive, higher proliferation potential of the resulting primary tumors and a higher rate of metastasis [15]. This suggests that our observation of platelet condensation of ovarian cancer spheroids in vitro might be relevant to increased aggressiveness of ovarian cancer in vivo. Thus, a greater understanding of how platelets alter ovarian cancer spheroid morphology is needed to develop approaches to prevent this functional interaction.

In the current study, we explored how platelets affect ovarian cancer spheroids using a mass spectrometry approach to identify proteins and pathways associated with these interactions. The liver X receptor/retinoid X receptor (LXR/RXR) and integrin signaling pathways were identified as the top pathways associated with platelet effects on ovarian cancer spheroid morphology. We validated that platelets inhibited LXR transactivation activity in cancer cells in a platelet number-dependent manner.

## 2. Materials and Methods

### 2.1. Collection and Processing of Platelets

Platelets were either acquired from the Oklahoma Blood Institute or collected from volunteers who had given their informed consent and were selected according to the IRB #1426. To obtain venous blood, 21G butterfly needles (BD 368656) were used, and the blood was collected in acid citrate dextrose (BD, Franklin Lakes, NJ, USA, 364606) anticoagulant. The VetScan HM5 Hematology Analyzer’s (Zoetis, Parsippany, NJ, USA) primate species function was used to perform a complete blood count (CBC) analysis. The platelet-rich plasma (PRP) was separated from the red and white blood cells by centrifugation at 1000 rpm for 10 min at room temperature without brakes. The platelet count in the PRP was then determined using the Zoetis VetScan HM5 Hematology Analyzer. The platelet-free plasma (PFP) control was obtained by centrifuging the blood at 800 G for 10 min at room temperature and collecting the upper fraction supernatant.

### 2.2. Cell Lines

Human ES-2/GFP-luciferase (hereafter referred to as ES-2) and MES-OV/GFP-luciferase (hereafter referred to as MES-OV) were generated as described before by Moisan and colleagues [16] and gifted by Dr. Branimir I Sikic at Stanford University. Both ES-2 and MES-OV cells were cultured in DMEM/high glucose media supplemented with 10% fetal bovine albumin and 1× penicillin/streptomycin solution. Human ES-2 (non-luciferase expressing), purchased from American Type Cell Culture, were a gift from Dr. Antony Burgett’s lab at the University of Oklahoma Health Sciences Center and were cultured in McCoy’s 5A media supplemented with 10% fetal bovine albumin and 1× penicillin/streptomycin solution.

### 2.3. Metabolic Viability Assays

A reduction assay using (3-(4,5-dimethylthiazol-2-yl)-2,5-diphenyltetrazolium bromide) tetrazolium (MTT) was used to determine cell viability as previously described [17]. Briefly, cancer cells that would be 60% confluent after an overnight incubation were plated in a 96-well plate overnight to allow attachment. The next day, increasing concentrations of platelet inhibitors, or matching concentrations of the control vehicle, were added to the cells containing wells in triplicate for 24 or 72 h. The platelet inhibitors used were aspirin (Cayman Chemical Company, Ann Arbor, MI, USA, 70260), celecoxib (Cayman Chemical Company, 10008672), clopidogrel (Cayman Chemical Company, 17871), dipyridamole (Cayman Chemical Company, 18189), eptifibatide (Cayman Chemical Company, 21,578 or MuseChem 188627-80-7), and prostacyclin (Cayman Chemical Company, 18220). After incubation, an MTT (CellTiter 96 Aqueous Non-Radioactive Cell Proliferation Assay, Promega, Madison, WI, USA, G4100) was used according to the manufacturer’s instructions. The optical density (OD) was measured at wavelengths of 570 and 620 nm using a Synergy H1 microplate reader (Agilent, Santa Clara, CA, USA). Percent viability was derived by dividing the OD of the treated cultures by the average OD of the cultures treated with solvent only.

### 2.4. Generation of In Vitro Spheroid Cultures

A 3D magnetic spheroid assay from Greiner Bio-One was used to generate a single spheroid per well according to the manufacturer’s instructions. Briefly, 60% confluent cells were magnetically sensitized by co-incubating with NanoShuttle™ Biocompatible nanoparticles(Greiner Bio-One, Kremsmünster, Austria, 657852) overnight prior to forming the spheres. Once cells were confluent, unbound nanoparticles were washed off and cells were cultured in culture media on an ultra-low attachment 96-well plate (Greiner Bio-One, 655976) on top of a magnetic spheroid drive holder to facilitate the formation of cell aggregates for one to four hours. The magnet was removed, and the resulting spheroids were used for further studies.

A U-shaped ultra-low attachment plate method was used as an alternative to generate organoids without magnetics. Briefly, about 80% confluent cells were trypsinized and harvested, washed, and resuspended in regular culture media supplemented with 10% fetal bovine albumin and 1× streptomycin/penicillin solution. An optimized number of cells was cultured on U-shaped ultra-low attachment 96-well plates (Greiner Bio-One 650,970 or Corning, Corning, NY, USA, 7007) in cell culture media, and the content was incubated for an optimized amount of time before sphere manipulation.

After successful formation of stable spheroids, platelet inhibitors or control vehicle were added to the spheroid-containing wells, followed by platelet rich plasma (PRP, 1 million platelets total) or platelet free plasma (PFP) control, and incubated for an optimized amount for time before imaging or protein isolation. The spheroids’ size and density were measured using the GelCount (Oxford Optronix, Oxford, UK).

### 2.5. Protein Isolation

Mammalian Protein Extraction Reagent (MPER, Thermo Fisher, Waltham, MA, USA, 78501) was used to extract proteins from cell lysates or spheroids according to the manufacturer’s instructions. Soluble proteins were extracted from conditioned media using size exclusion methods (EMD Millipore, Burlington, MA, USA, UFC500396) according to the manufacturer’s instructions. Protein concentrations for each sample were estimated using the Pierce BCA Protein Assay (Fisher Scientific, Waltham, MA, USA, PI23225), according to the manufacturer’s instructions.

### 2.6. Mass Spectrometry

Mass spectrometry was performed as we previously published [18]. In brief, the FASP protocol [19] was used to process proteins (10 µg) isolated from each sample. Proteins were digested with 0.2 µg Promega Sequencing Grade Modified Trypsin (0.2 µg, Promega, Madison, WI, USA, V5111) overnight at 37 °C in 40 mM NH_4_HCO_3_. A control sample containing an equal amount of each individual sample was prepared and subjected to the same procedure to ensure robustness of the data. The digested samples were desalted, dried, and resuspended in 100 mM triethylamine ammonium bicarbonate before labeling the proteins with TMT-11plex (Thermo Fischer Scientific, Waltham, MA, USA). The TMT-11 channel was used for the “mix” sample, as a reference for the different TMT runs.

Pierce C 18 spin columns (Thermo Fischer Scientific, Waltham, MA, USA) were used to desalt and concentrate the TMT-labeled tryptic peptides. A ThermoFisher Acclaim PepMap 100 C18 sequencing column was loaded with 1 ug of the tryptic peptides, which were then quantified after a 120-min acetonitrile gradient elution. Liquid chromatography tandem mass spectrometry (LC-MS/MS) with a Thermo Lumos Fusion tribrid Orbitrap mass spectrometer coupled to an Ultimate 3000 RSLC nano ultra-high-performance liquid chromatography (UHPLC) was used to analyze the eluted peptides.

To identify proteins, the search engine SEQUEST and Proteome Discoverer 2.4 with the human Uniprot proteome database version 20,201,123 (42,412 reviewed proteins) as a reference were utilized. At least two peptides per protein were required for protein identification. A false discovery rate (FDR) of ≤1%, a fragment ion mass tolerance of 0.05 Da, a precursor ion mass tolerance of 10 ppm, and three missed tryptic cleavage sites were used to restrict the search parameters in the database. N-terminal acetylation (+42) and Met oxidation (+16 Da) were used as variable protein modifications, while Cys carboxymethylation (+58 Da) was the fixed protein modification. Overlapping isotope contributions from the TMT tags in the reporter ion intensities were bias-corrected using information in the manufacturer’s certificate. General recommended guidelines were utilized in the reporting of proteins, i.e., protein FDR cutoff ≥1%, ≥2 peptide matches, ≥7 amino acid minimum peptide length, and 3 biological replicates. For analysis, proteins with a “High” value for “FDR Confidence Combined” were utilized, while all other parameters were set to default. To equalize the amounts of protein labeled by each TMT reagent, total signals in each of the channels were corrected with computed normalization factors. Data were exported to Microsoft Excel 2019 and Perseus (version 1.6.15.0) to conduct the subsequent analysis.

### 2.7. Generation of LXRα Inducible Luciferase Reporter ES-2 Cells

We established an LXRα inducible luciferase ES-2 cell line by stably transducing the pGreenFire1-LXRE-in-LXRα Lentivector (System Biosciences, Palo Alto, CA, USA, TR102VA-P) into the ES-2 (non-luciferase expressing original cells) ovarian cancer cell line. Briefly, ES-2 cells were transduced with the lentivector and selected with Puromycin (1 µg/mL) until all the control, non-transduced cells were completely dead. LXR expression and activation was confirmed by measuring luciferase activity in transduced ES-2 cells after adding an LXR agonist, T0901317 or LXR antagonist, GSK2033 (Cayman Chemical, 71810, or 25443, respectively). Luciferase activity was measured after adding the luciferin substrate (PerkinElmer, Waltham, MA, USA, 122799) with the In Vivo Imaging System (IVIS, PerkinElmer, Waltham, MA, USA). Nonlinear regression analysis was conducted using Prism 9.4 (GraphPad, La Jolla, CA, USA) to validate an increase in luminescence as an indirect measure of LXR activation.

### 2.8. Statistical and Bioinformatic Analysis

Prism 9.4 (GraphPad) was used to first determine if data were distributed normally, and then to evaluate two samples with a *t*-test or Mann–Whitney test, or ANOVA or Kruskal–Wallis test, depending on whether the data was normally distributed or not, respectively. *p* values of <0.05 were considered statistically significant. Proteins identified in mass spectrometry analysis were determined to be significantly altered between experimental conditions using volcano plots with *p* values < 0.05 or −log10 > 1.3 in combination with at least two-fold differences (log2 ≥ 1 or log2 ≤ 1), used for the cutoffs. Ingenuity pathway analysis (IPA, Qiagen, Hilden, Germany) was used to identify pathways affected by the significantly altered proteins.

## 3. Results

### 3.1. Platelets, but Not Platelet Inhibitors, Altered Ovarian Cancer Spheroids Properties

We used our 3D magnetic ovarian cancer spheroid-platelet co-culture model [14] to test platelet-mediated effects on ovarian cancer cells grown as spheroids. To validate these effects of platelets on spheroid characteristics, we identified US Food and Drug Administration (FDA)-approved drugs that could interfere with this process and be repurposed for preventing platelets from causing thrombosis and increasing cancer aggressiveness. We utilized multiple classes of platelet inhibitors with different mechanisms of action and identified achievable plasma concentration ranges when administered to humans. The compounds chosen were aspirin, celecoxib, clopidogrel, dipyridamole, eptifibatide, and prostacyclin. Their mechanisms of action and achievable plasma concentrations are listed in Table 1.

To rule out the direct effect of these compounds on epithelial cells in our cell culture models, a cytotoxicity assay using MTT to measure metabolic viability was performed both at 24 and 72 h of treatment time in ES-2, a clear cell carcinoma cell line, and MES-OV, a high grade serous ovarian cancer cell line, to establish doses that did not affect epithelial cells (Figure 1 and Appendix A).

Therefore, when used in our co-culture experiments, any effect that we observed at treatment concentrations below the minimal concentration found to cause reduction in epithelial viability was assumed to be platelet-mediated. To evaluate the potential effects of these antiplatelet drugs on platelet-mediated spheroid changes, ovarian cancer spheroids in culture were treated with platelet inhibitors in the presence or absence of platelets. The findings revealed that most platelet inhibitors were able to reverse platelet-mediated effects on ovarian cancer spheroids (Figure 2, Appendix A).

Aspirin preferentially targets cyclooxygenase (COX)-1, while celecoxib selectively inhibits COX-2; hence, they preferentially mediate formation of series-1 or series-2 prostaglandins, respectively. Both COX inhibitors reversed platelet-mediated alterations in the sizes and densities of ovarian cancer spheroids, suggesting involvement of series-1 and series-2 in the functional interactions of platelets and ovarian cancer spheroids. Additionally, eptifibatide and prostacyclin (PGI2), which target platelet glycoprotein GPIIB-IIIA binding, and platelet’s G-protein coupled receptor activation, respectively, inhibited the platelets’ effect on ovarian cancer spheroids, suggesting the involvement of these receptors also in the functional interactions of platelets and ovarian cancer spheroids. On the other hand, clopidogrel and dipyridamole, which target the adenosine signaling pathway, were unable to reverse platelet effects on ovarian cancer spheroids, hence ruling out the importance of adenosine in platelets and tumor interactions.

Intriguingly, eptifibatide, the GPIIB-IIIA integrin inhibitor, and dipyridamole, an adenosine reuptake inhibitor, showed opposite effects on the spheroids. While eptifibatide inhibited platelet-mediated compression of the spheroids, dipyridamole enhanced the platelet-mediated compression, suggesting that the two drugs would have opposite effects if used in cancer treatment. To rule out any direct effects of these drugs on epithelial cells, the effects of dipyridamole and eptifibatide on cancer spheroids were tested in the absence of platelets. Neither drug caused any changes to spheroid sizes or densities in the absence of platelets (Figure 3), hence confirming that the observed spheroid size and density alterations were platelet-mediated, and that the anti-platelet drugs exerted their effects on the spheroids indirectly through the platelets, and not by directly affecting the cancer cells in the spheroids.

Ovarian cancer patients with thrombocytosis have reduced progression-free and overall survival [26,27]. Platelets have been shown to have ovarian cancer tumor-promoting effects in vivo and in vitro [11]. In our model, co-incubation of 2D cultures of ovarian cancer cells grown with platelet-rich plasma from non-cancer patients did not have any effect on cancer cell metabolic viability (Appendix A). Hence, we developed a 3D magnetic ovarian cancer spheroid-platelet co-culture model to test platelet-mediated effects on ovarian cancer cells grown as spheroids [14]. The measurement of digital images of the spheroids documented that the platelet co-incubation resulted in significantly smaller and denser spheroids (Figure 2 and Figure 3).

### 3.2. Ovarian Cancer Spheroids Co-Incubation with Platelets Affected the LXR and Integrin Signaling Pathways

To better understand the pathways that were being affected by direct interaction between ovarian cancer spheroids and platelets, mass spectrometry analysis was performed on proteins collected from magnetically generated spheroids and conditioned media after incubation with platelets or no platelets for 90 min. Two independent mass spectrometry analyses (n = 3, platelet specimens from three individual healthy volunteers for each experiment) were conducted for proteins in both cell lysates and conditioned media (Appendix A). Volcano plots were used to identify proteins that were significantly different in the samples from cultures incubated in the presence or absence of platelets (Appendix A).

The pathway most activated by platelet exposure was identified by IPA analysis to be the liver X receptor/retinoid X receptor (LXR/RXR) signaling pathway in the conditioned media that resulted from co-incubation of ovarian cancer spheroids and platelets for both independent experiments, followed by acute phase response signaling (Figure 4A, Appendix A). FXR/RXR pathway activation also was identified; however, there was no activity pattern with this pathway. The most affected pathways in the protein lysates of the spheroids were the LXR/RXR and the integrin pathways (Figure 4B).

IPA identified the three platelet receptors, integrin alpha-IIb, integrin beta 3, and the dimeric complex of both receptors GPIIB-IIIA, in the interaction network of molecules altered by the platelets (Figure 5). These receptors are known upstream effectors of the LXR-RXR molecules altered by platelet co-incubation with cancer cells. Then, we added eptifibatide to the network analysis, which verified that it directly inhibits the three identified integrin receptors (Figure 5).

### 3.3. Validation of LXR/RXR Signaling Pathways in the Platelet-Ovarian Cancer Spheroids Interaction

To validate the involvement of the LXR-RXR pathway in the platelet/cancer interaction, we established and utilized an inducible LXR-GFP luciferase-expressing reporter ES-2 cell line to measure platelet effects on LXR transactivation activity. Treatment of these reporter cells with an LXR agonist or LXR antagonist at their half-maximal inhibitory concentrations increased or decreased luciferase activity, respectively, in the cells, validating the functional activity of LXR in this reporter line (Figure 6).

Then, we co-incubated spheroids of this ES-2 reporter cell line with increasing numbers of platelets and observed a gradual reduction in luciferase activity with an increasing number of co-incubated platelets indicating that the platelets inhibited LXR activity in ES-2 cells (Figure 6). To further validate these results, we measured platelet effects on luciferase activity in the presence of an LXR antagonist or agonist. To rule out that this activity was not due to plasma instead of platelets, we treated spheroids with the same amount of PFP and found no change in LXR activity when plasma was added to spheroids.

## 4. Discussion

In this study, we utilized our previously published 3D co-culture model [14] to study how platelets alter the size and density of ovarian cancer spheroids as a model of ovarian cancer spheroids present in ascites. Previously, we ruled out the effects of dihomo-gamma-linolenic acid in this mechanism [14]. In this study, integrin and LXR/RXR signaling were identified to be upregulated in platelet and ovarian cancer cell co-cultures. Evaluation of a series of platelet inhibitors at physiologically achievable concentrations that were below their half-maximal inhibitory concentrations for ovarian cancer cells identified that eptifibatide interfered with, while dipyridamole enhanced, the effects of platelets on ovarian cancer spheroids.

Eptifibatide inhibits platelet aggregation and limits bleeding risk in patients [28], and is FDA-approved for treatment of acute coronary syndrome and percutaneous coronary intervention [29]. Our findings and other published studies support the repurposing of eptifibatide as an anti-cancer agent. Eptifibatide has been shown to prevent cell adhesion, migration, invasion, and proliferation of ovarian, kidney, and cervical cancer cells [30,31]. In breast cancer models, eptifibatide induced apoptosis [32], inhibited platelet-mediated cell invasion [33], and reduced epithelial-to-mesenchymal transition markers [34]. In a mastocytoma model, eptifibatide inhibited prostaglandin E2-stimulated adhesion of cancer cells to the Arg-Gly-Asp-enriched matrix [35]. In a lung cancer model, eptifibatide inhibited platelet-mediated increase in PD-L1 on A549 cells and inactivation of T cells [36]. In the current study, eptifibatide did not show direct anticancer effect in the absence of platelets. In summary, our findings and findings by others show that the eptifibatide anticancer activity is highly dependent on the presence of platelets, hence thrombocytosis would be an appropriate screening factor to identify patients who would most likely benefit from use of eptifibatide as an anti-cancer therapeutic.

Eptifibatide functions by interfering with ligand binding to the GPIIB-IIIA integrin receptor [28]. GPIIB-IIIA is mainly expressed on activated platelets and is also expressed on cells present in ascites fluid collected from patients with ovarian cancer, while being scarcely expressed on tumor cells of ovarian and renal cancers [30,37]. In renal and ovarian cancers, GPIIB-IIIA acts as a precursor to α2β1 integrin-mediated adhesion and invasion when in crosstalk with cadherin 6 [30]. Tirofiban, another GPIIB-IIIA inhibitor, has been shown to inhibit chemotaxis of squamous cell carcinoma [38] and work synergistically with cisplatin to boost chemotherapy effects [39]. Taken together, these data support targeting GPIIB-IIIA in the development of cancer therapeutics.

Dipyridamole, on the other hand, represents a drug that should be used with caution, or not at all in patients with ovarian cancer. Dipyridamole is an FDA-approved adjunctive drug for thromboembolism prophylaxis in patients undergoing cardiac valve replacement and thallium-nuclear stress testing [40]. Our observed promotion of platelet effects on ovarian cancer spheroids by dipyridamole provides an opportunity to better understand platelet promotion in ovarian cancer. Dipyridamole functions as an anti-platelet drug by inhibiting adenosine reuptake. Previous observations of dipyridamole anti-cancer activity have occurred mainly through enhancement by, or synergy with, other treatments. Cell culture models have shown that dipyridamole increases the sensitivities of cancer cells to interferon [41], tumor necrosis factor α (TNFα) [42], and cisplatin [43]. Furthermore, dipyridamole has been shown to enhance adenosine-induced cisplatin toxicity towards cancer cells [43]. In a phase one clinical trial, dipyridamole enhanced etoposide anti-cancer activity [44]. In our study, dipyridamole did not have any direct effects on ovarian cancer in the absence of platelets. Further research is needed to better understand the interactions of dipyridamole with other anti-cancer drugs, both in the presence and in the absence of platelets.

Both eptifibatide and dipyridamole are used as blood thinners, which generally increase the bleeding risk in patients. Other commonly reported side effects of dipyridamole include dizziness, headache, chest pain, angina exacerbation, and abnormal electrocardiogram [40]. A phase one clinical trial combining etoposide and dipyridamole reported myelosuppression as the dose-limiting toxicity, while other symptoms like nausea and vomiting were mild [44]. Eptifibatide’s additional side effects include thrombocytopenia; however, eptifibatide-induced thrombocytopenia is less common compared to what has been observed with other drugs in the same class, such as abciximab and tirofiban [29]. Ovarian cancer patients have an increased risk of blood clotting [45,46], and this risk is further exacerbated by chemotherapy [47], which supports the use of blood thinners, such as eptifibatide and dipyridamole, as a potential strategy to prevent unwanted blood clots in cancer patients.

We identified that platelet co-incubation with ovarian cancer spheroids altered the LXR/RXR signaling pathway. Our bioinformatic analysis suggests that the altered LXR/RXR signaling is likely a downstream consequence of GPIIB-IIIA and other integrin activation. There are two LXRs (LXRα and LXRβ) and three RXRs (RXRα, RXRβ and RXRγ) receptors, which can form multiple heterodimers with different functions. The heterodimers transactivate target genes by binding to the LXR response element (LXRE). The ligand-dependent LXR/RXR heterodimer is activated to function as a transcription factor by binding of natural and synthetic oxysterol ligands to the LXR partner, or *9-cis*-retinoic acid or synthetic rexinoids to the RXR partner [48]. While the LXR partner requires binding to an RXR partner to function as a transcription factor, the RXRs can function as homodimers and as binding partners to several other nuclear receptors [49].

The LXR/RXR heterodimers and their agonists regulate multiple cancers both in vitro and in vivo [50,51]. LXR agonists have been observed to induce G1/S phase cell cycle arrest in prostate, lung, breast, liver, cervical, colorectal, and ovarian cancer cells [52,53,54,55,56,57], and reduce tumor growth in breast, prostate, liver, and lung cancer xenograft models [52,53,56,58,59,60] using multiple mechanisms. In colon cancer, LXR agonists suppressed carcinogenesis by blocking the activation of Wingless/Integrated signaling, decreasing expression of the β-catenin target genes and inducing pyroptosis [51]. In breast cancer models, LXR agonists inhibited estrogen regulation of hepatic estrogen sulfotransferase in estrogen-dependent cancer cells [51].

The potential anti-cancer effects of LXR agonists include the inhibition of angiogenesis. Studies of human umbilical vein endothelial cells have demonstrated that LXR agonists inhibit tubulogenesis via their interference with cholesterol homeostasis, which disrupts lipid raft localization and signaling of vascular endothelial growth factor receptor-2 (VEGFR-2) [61]. LXR agonists could also inhibit angiogenesis indirectly through their effects on platelets. The extensive role of platelets in promoting tumor angiogenesis has been reported by Filipelli et. al. [62]. In their review, the authors compiled preclinical and clinical evidence showing that platelet release of their granule contents, including VEGF, platelet derived growth factor (PDGF), fibroblast growth factor-2 (FGF-2), and matrix metalloproteases (MMPs) increases angiogenesis by at least 25%, which in turn increases the proliferation, migration, and invasiveness, and reduces the cisplatin sensitivity, of tumor cells. In murine models, the depletion of platelets has been correlated with decreased angiogenesis, as measured by decreased microvessel density in tumors. Furthermore, clinical studies have demonstrated that an elevated platelet count is correlated with advanced disease stage and levels of angiogenic factors including VEGF, PDGF, interleukin-6, thrombospondin, platelet factor 4, and transforming growth factor β [62].

The RXR receptor and its ligands play multiple roles in cancer models of melanoma, lung, kidney, gastric, liver, breast, cervical, and prostate cancers [63,64,65,66,67,68,69,70,71,72,73]. RXR inhibitors inhibit the AKT pathway, which results in apoptosis, prevention of M2 macrophage polarization, delayed cancer relapse, reduced drug resistance, and mitotic catastrophe [74]. Bexarotene is an RXR agonist that is FDA-approved and indicated for cutaneous T-cell lymphoma, prescribed off-label for non-small cell lung cancer, and has been shown to have anti-cancer activity in breast and colon cancer models [75,76]. In ovarian cancer, bexarotene induces pyroptosis, depletes total glutathione, represses nuclear factor erythroid 2-related factor 2-dependent transcription, elevates cellular reactive oxygen species, and inhibits protein kinase B phosphorylation [77,78], which further supports the RXR/LXR signaling pathway as a potential therapeutic target in ovarian cancer.

## 5. Conclusions

This study identified the integrin and LXR/RXR signaling pathways as candidate targets for the development of improved strategies to treat patients with ovarian cancer, specifically by interfering with platelet promotion of ovarian cancer and complementing the activities of other anti-cancer drugs. Further study on repurposing eptifibatide to treat ovarian cancer by interfering with the GPIIB-IIIA integrin receptor on platelets with downstream modulation of LXR/RXR signaling is justified. Additional studies of dipyridamole for its potential promotion of ovarian cancer by inhibiting adenosine reuptake in platelets is also warranted to develop recommendations surrounding the use of this drug and other adenosine reuptake inhibitors in patients with ovarian cancer.

## Figures and Tables

**Figure 1 cancers-16-03533-f001:**
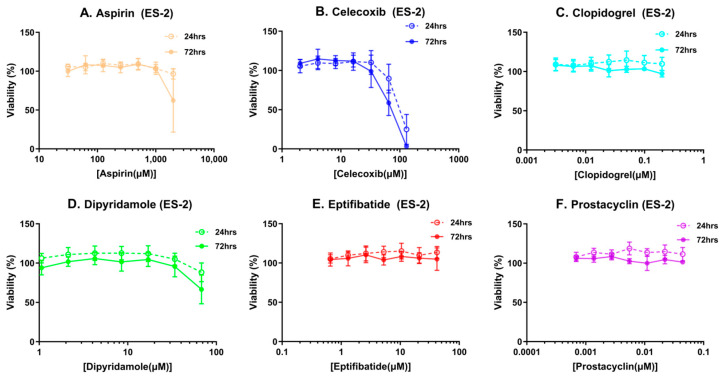
(**A**–**F**) Cytotoxicity profiles of platelet inhibitors in ES-2 ovarian cancer cells: MTT assay was used to determine cell viability at 24 and 72 h post-treatment with platelet inhibitors at various concentrations.

**Figure 2 cancers-16-03533-f002:**
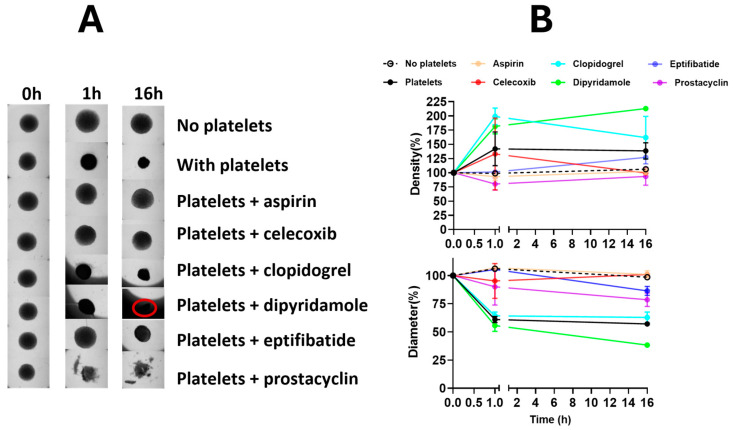
Platelet inhibitors modulate ovarian cancer spheroids in the presence of platelets. (**A**) Spheroid images. (**B**) Spheroid density and diameter quantification. Magnetic ES-2 spheroids were co-incubated without platelets for control or with platelets and platelet inhibitors before imaging with a GelCount to derive diameter and density values for each spheroid. *p*-values < 0.05 were considered significant. Dipyridamole-treated spheroids were overshadowed by the wall of the well at 16 h because dipyridamole caused spheroids to migrate to the side of the wall, therefore we outlined the spheroid with a red circle.

**Figure 3 cancers-16-03533-f003:**
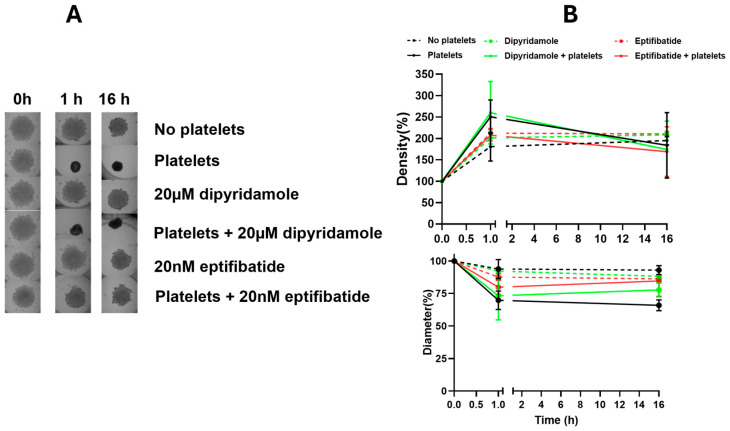
Effect of dipyridamole and eptifibatide without platelets. (**A**) Spheroid images. (**B**) Spheroid density and diameter quantification. Magnetic ES-2 spheroids were co-incubated with platelets for control, or without platelets and platelet inhibitors, before imaging with a GelCount to derive diameter and density values for each spheroid. *p*-values < 0.05 were considered significant.

**Figure 4 cancers-16-03533-f004:**
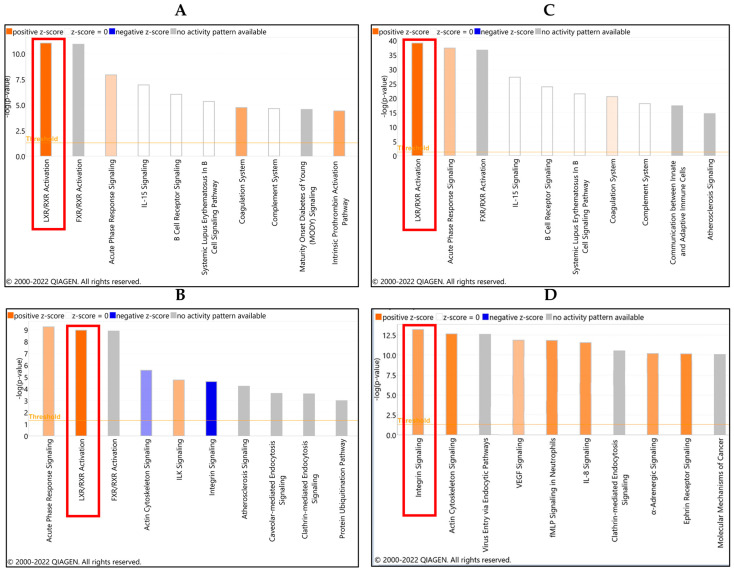
(**A**,**B**) Canonical pathways affected by platelets in ovarian cancer spheroids-conditioned media. All molecules identified by mass spectrometry analysis that were altered beyond two-fold changes at a significant level were analyzed using IPA to identify top canonical pathways affected in the platelet/ovarian cancer spheroid co-culture conditioned media. (**A**) Experiment #1, (**B**) Experiment #2. (**C**,**D**) Canonical pathways affected by platelets in ovarian cancer spheroids. All molecules identified by mass spectrometry analysis that were altered beyond two-fold changes at a significant level were analyzed using IPA to identify top canonical pathways affected in the platelet/ovarian cancer spheroid co-culture cell lysates. (**C**) Experiment #1, (**D**) Experiment #2. Red rectangles indicate the LXR/RXR activation and integrin signaling pathways. The legend at the top of each panel shows which color indicates each activity pattern direction, with darker colors indicating higher z-scores and lighter colors indicating lower z-scores.

**Figure 5 cancers-16-03533-f005:**
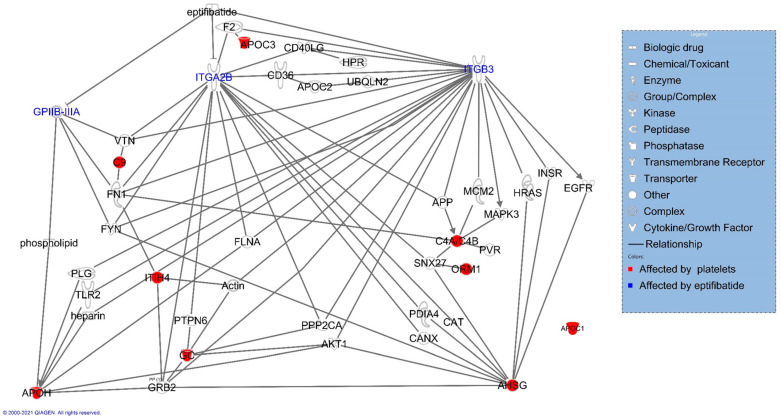
Association of platelet-altered molecules with known eptifibatide activity. IPA analysis of significantly altered genes demonstrated their connection with eptifibatide using the pathway connect function.

**Figure 6 cancers-16-03533-f006:**
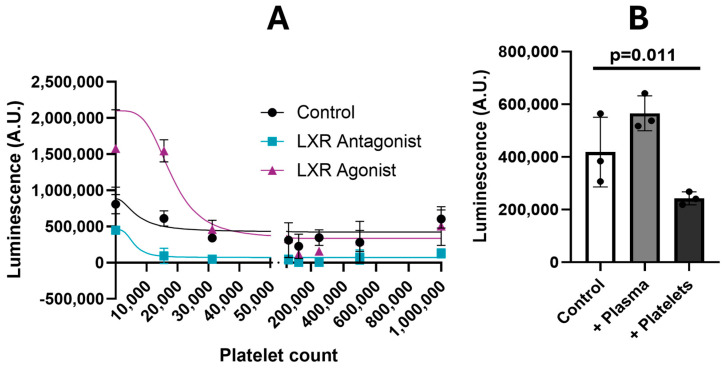
Effect of platelets on LXR activity. Spheroids were co-incubated with platelets or plasma control for 90 min in the presence of an LXR agonist or LXR antagonist. Luciferase activity was used to determine LXR activity. (**A**) Effects of the indicated amounts of platelets on LUC reporter activity in ES-2 cells in the presence and absence of an LXR antagonist or agonist. (**B**) Comparison of effects of plasma and 1 million platelets on LUC reporter activity in ES-2 cells. A.U. = absorbance units.

**Table 1 cancers-16-03533-t001:** Platelet inhibitors’ achievable plasma concentration: Values were derived from the literature-reported pharmacokinetics profiles for each drug.

Compound Name	Mechanism of Action	Achievable Plasma Concentrations (µM)	Reference
Aspirin	Cyclooxygenase 1 inhibitor	555	[20]
Celecoxib	Cyclooxygenase 2 inhibitor	64.37	[21]
Clopidogrel	Inhibits adenosine disphosphate binding to its purinergic receptor P2Y12	0.099	[22]
Dipyridamole	Inhibits the reuptake of adenosine into platelets	33.69	[23]
Eptifibatide	Reversible binding of glycoprotein IIB/IIIA	26.44	[24]
Prostacyclin *	Platelet aggregation inhibitor	0.11	[25]

* Epoprostenol, major prostacyclin metabolite concentration. Epoprostenol was used because the half-life of prostacyclin is only 42 s.

## Data Availability

Data generated in this manuscript are included in the Appendix A.

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
