# Peer review of "Implications of GPIIB-IIIA Integrin and Liver X Receptor in Platelet-Induced Compression of Ovarian Cancer Multi-Cellular Spheroids"

_cancers, 2024, doi:10.3390/cancers16203533_

Round 1
Reviewer 1 Report
Comments and Suggestions for Authors
Dear authors,
I read with interest your manuscript, which I found to be of good quality. I have just one comment regarding the introduction/discussion sections, which I think should be implemented
The authors should introduce and discuss better the role that platelets can have in promoting the angiogenesis of the tumor, which is only partially reported in the introduction section of the manuscript (please see PMID: 36362186). This is very important since the discussion section suggests that LRX therapy can also reduce angiogenesis through the inhibition of vascular endothelial growth factor which is synthesized by platelets.
Author Response
I read with interest your manuscript, which I found to be of good quality. I have just one comment regarding the introduction/discussion sections, which I think should be implemented
The authors should introduce and discuss better the role that platelets can have in promoting the angiogenesis of the tumor, which is only partially reported in the introduction section of the manuscript (please see PMID: 36362186). This is very important since the discussion section suggests that LRX therapy can also reduce angiogenesis through the inhibition of vascular endothelial growth factor which is synthesized by platelets.
The role of platelets-derived VEGF has been added in the introduction (lines 60-62 ) and discussion sections (lines 435 to 454 in the marked up version of revision).
Reviewer 2 Report
Comments and Suggestions for Authors
The paper is well-organized with a strong structure and insightful discussion, making it a promising candidate for publication. However, certain areas could be improved to enhance the overall quality and clarity of the manuscript.
One of the key findings, that platelets—but not platelet inhibitors—alter ovarian cancer spheroid properties, is both unique and noteworthy. This discovery adds a new dimension to understanding the interaction between platelets and tumor dynamics. However, to strengthen the validity of this finding, it is recommended that additional data be provided. These data could include expanded experimental replicates or further mechanistic insights to enhance the accuracy and reliability of the conclusions drawn.
In Figure 1, the cytotoxicity profiles of the tested compounds are presented, yet the concentration unit for eptifibatide is missing, which could lead to ambiguity for readers. To improve clarity, it is crucial to include this missing information. Furthermore, adding labels such as "A, B, C" to distinguish the different panels would enhance readability and allow for easier cross-referencing in the discussion. It is also advisable to apply this labeling system consistently throughout the manuscript to maintain uniformity across all figures, ensuring that readers can easily follow the narrative and interpret the data presented.
For Figure 2, while the spheroid results are visually compelling, the absence of a scale bar detracts from the precision of the data. Including a scale bar is essential for accurate interpretation of the spheroid dimensions and any morphological changes being studied. Moreover, it is suggested to revisit the overall layout of the figure to make it more user-friendly. Increasing the size of the tick marks on the axes and adjusting the aspect ratio of the plots would further enhance the figure’s readability, allowing the data to be more easily interpreted by the audience.
Similarly, for Figure 4, the figures could benefit from an increase in size. Enlarging the figures and optimizing the layout would not only improve their visibility but also make the underlying data more accessible to readers. In doing so, the visual presentation of the research would align more closely with the clarity and depth of the written discussion, creating a more cohesive and reader-friendly experience overall.
For Figure 6, it is recommended that panels A and B follow the same formatting for consistency and clarity. Ensuring that both panels are presented in the same format will make it easier for readers to compare and interpret the data. This includes aligning the axis scales, maintaining uniform font sizes for labels, and using the same color schemes or line styles where applicable. By harmonizing the formatting of both panels, the figure will appear more cohesive and professional, improving the overall presentation of the results.
Comments on the Quality of English LanguageMinor revision are needed
Author Response
The paper is well-organized with a strong structure and insightful discussion, making it a promising candidate for publication. However, certain areas could be improved to enhance the overall quality and clarity of the manuscript.
One of the key findings, that platelets—but not platelet inhibitors—alter ovarian cancer spheroid properties, is both unique and noteworthy. This discovery adds a new dimension to understanding the interaction between platelets and tumor dynamics. However, to strengthen the validity of this finding, it is recommended that additional data be provided. These data could include expanded experimental replicates or further mechanistic insights to enhance the accuracy and reliability of the conclusions drawn.
We agree with this statement. Our future studies will expand this study using other biological and experimental replicates and expand it in other cancers. We request reviewer 2 to consider this manuscript as a significant contribution to science with sufficiently robust experiments and allow us to publish the results at this point. This publication will support our grant applications to continue the research, gain further mechanistic insight and then translate the research toward clinical trials.
In Figure 1, the cytotoxicity profiles of the tested compounds are presented, yet the concentration unit for eptifibatide is missing, which could lead to ambiguity for readers. To improve clarity, it is crucial to include this missing information. Furthermore, adding labels such as "A, B, C" to distinguish the different panels would enhance readability and allow for easier cross-referencing in the discussion. It is also advisable to apply this labeling system consistently throughout the manuscript to maintain uniformity across all figures, ensuring that readers can easily follow the narrative and interpret the data presented.
The figure has been updated with the correct concentration of eptifibatide and the panel labels have been added.
For Figure 2, while the spheroid results are visually compelling, the absence of a scale bar detracts from the precision of the data. Including a scale bar is essential for accurate interpretation of the spheroid dimensions and any morphological changes being studied. Moreover, it is suggested to revisit the overall layout of the figure to make it more user-friendly. Increasing the size of the tick marks on the axes and adjusting the aspect ratio of the plots would further enhance the figure’s readability, allowing the data to be more easily interpreted by the audience.
Figure 2 has been updated and adjusted to make it more user friendly. Unfortunately, adding a scale bar with the instrument that was used to quantify the spheroids is challenging, hence a supplementary table showing the row data before analysis or normalization has been added.
Similarly, for Figure 4, the figures could benefit from an increase in size. Enlarging the figures and optimizing the layout would not only improve their visibility but also make the underlying data more accessible to readers. In doing so, the visual presentation of the research would align more closely with the clarity and depth of the written discussion, creating a more cohesive and reader-friendly experience overall.
Figure 4 has been enlarged to improve readibility.
For Figure 6, it is recommended that panels A and B follow the same formatting for consistency and clarity. Ensuring that both panels are presented in the same format will make it easier for readers to compare and interpret the data. This includes aligning the axis scales, maintaining uniform font sizes for labels, and using the same color schemes or line styles where applicable. By harmonizing the formatting of both panels, the figure will appear more cohesive and professional, improving the overall presentation of the results.
Figure 6 has been updated using the same formatting style for both panels. Axis scales have been aligned at the bottom and labels have been adjusted to the same size. The colors have been adjusted in panel B.